# The Convergence of Sparsified Gradient Methods

**Dan Alistarh**[*]
IST Austria
dan.alistarh@ist.ac.at

**Torsten Hoefler**
ETH Zurich
htor@inf.ethz.ch

**Mikael Johansson**
KTH
mikaelj@kth.se

**Sarit Khirirat**
KTH
sarit@kth.se

**Nikola Konstantinov**
IST Austria
nikola.konstantinov@ist.ac.at

**Cédric Renggli**
ETH Zurich
cedric.renggli@inf.ethz.ch

## Abstract

Stochastic Gradient Descent (SGD) has become the standard tool for distributed training of massive machine learning models, in particular deep neural networks. Several families of communication-reduction methods, such as quantization, large-batch methods, and gradient sparsification, have been proposed to reduce the overheads of distribution. To date, gradient sparsification methods–where each node sorts gradients by magnitude, and only communicates a subset of the components, accumulating the rest locally–are known to yield some of the largest practical gains. Such methods can reduce the amount of communication per step by up to *three orders of magnitude*, while preserving model accuracy. Yet, this family of methods currently has no theoretical justification.

This is the question we address in this paper. We prove that, under analytic assumptions, sparsifying gradients by magnitude with local error correction provides convergence guarantees, for both convex and non-convex smooth objectives, for data-parallel SGD. The main insight is that sparsification methods implicitly maintain bounds on the maximum impact of stale updates, thanks to selection by magnitude. Our analysis also reveals that these methods do require analytical conditions to converge well, justifying and complementing existing heuristics.

## 1 Introduction

The proliferation of massive datasets has led to renewed focus on distributed machine learning computation. In this context, tremendous effort has been dedicated to scaling the classic *stochastic gradient descent (SGD)* algorithm, the tool of choice for training a wide variety of machine learning models. In a nutshell, SGD works as follows. Given a function $f : \mathbb{R}^n \to \mathbb{R}$ to minimize and given access to stochastic gradients $\tilde{G}$ of this function, we apply the iteration

$$x_{t+1} = x_t - \alpha \tilde{G}(x_t), \tag{1}$$

where $x_t$ is our current set of parameters, and $\alpha$ is the step size.

The standard way to scale SGD to multiple nodes is via *data-parallelism*: given a set of $P$ nodes, we split the dataset into $P$ partitions. Nodes process samples in parallel, but each node maintains a globally consistent copy of the parameter vector $x_t$. In each iteration, each node computes a new stochastic gradient with respect to this parameter vector, based on its local data. Nodes then aggregate all of these gradients locally, and update their iterate to $x_{t+1}$. Ideally, this procedure would enable us to process $P$ times more samples per unit of time, equating to linear scalability.

---

[*]Authors ordered alphabetically. The full version can be found at https://arxiv.org/abs/1809.10505.

However, in practice scaling is limited by the fact that nodes have to exchange full gradients upon every iteration. To illustrate, when training a deep neural network such as AlexNet, each iteration takes a few milliseconds, upon which nodes need to communicate gradients in the order of 200 MB each, in an all-to-all fashion. This communication step can easily become the system bottleneck [4].

A tremendous amount of work has been dedicated to addressing this scalability problem, largely focusing on the data-parallel training of neural networks. One can classify proposed solutions into a) *lossless*, either based on factorization [31, 7] or on executing SGD with extremely large batches, e.g., [11], b) *quantization-based*, which reduce the precision of the gradients before communication, e.g., [22, 8, 4, 29], and c) *sparsification-based*, which reduce communication by only selecting an "important" sparse subset of the gradient components to broadcast at each step, and accumulating the rest locally, e.g., [24, 9, 2, 26, 17, 25].

While methods from the first two categories are efficient and provide theoretical guarantees, e.g., [31, 4], some of the largest benefits in practical settings are provided by sparsification methods. Recent work [2, 17] shows empirically that the amount of communication per node can be reduced by up to $600\times$ through sparsification *without loss of accuracy* in the context of large-scale neural networks. (We note however that these methods do require significant additional hyperparameter optimization.)

**Contribution.** We prove that, under analytic assumptions, gradient sparsification methods in fact provide convergence guarantees for SGD. We formally show this claim for both convex and non-convex smooth objectives, and derive non-trivial upper bounds on the convergence rate of these techniques in both settings. From the technical perspective, our analysis highlights connections between gradient sparsification methods and asynchronous gradient descent, and suggests that some of the heuristics developed to ensure good practical performance for these methods, such as learning rate tuning and gradient clipping, might in fact be *necessary* for convergence.

Sparsification methods generally work as follows. Given standard data-parallel SGD, in each iteration $t$, each node computes a local gradient $\tilde{G}$, based on its current view of the model. The node then *truncates* this gradient to its top $K$ components, sorted in decreasing order of magnitude, and accumulates the error resulting from this truncation locally in a vector $\epsilon$. This error is added to the current gradient *before* truncation. The top $K$ components selected by each node in this iteration are then exchanged among all nodes, and applied to generate the next version of the model.

Sparsification methods are reminiscent of *asynchronous* SGD algorithms, e.g., [20, 10, 8], as updates are not discarded, but delayed. A critical difference is that sparsification does not ensure that every update is eventually applied: a "small" update may in theory be delayed forever, since it is never selected due to its magnitude. Critically, this precludes the direct application of existing techniques for the analysis of asynchronous SGD, as they require bounds on the maximum delay, which may now be infinite. At the same time, sparsification could intuitively make better progress than an arbitrarily-delayed asynchronous method, since it applies $K$ "large" updates in every iteration, as opposed to an arbitrary subset in the case of asynchronous methods.

We resolve these conflicting intuitions, and show that in fact sparsification methods converge relatively fast. Our analysis yields new insight into this popular communication-reduction method, giving it a solid theoretical foundation, and suggests that prioritizing updates by magnitude might be a useful tactic in other forms of delayed SGD as well.

Our key finding is that this algorithm, which we call TopK SGD, behaves similarly to a variant of asynchronous SGD with "implicit" bounds on staleness, maintained seamlessly by the magnitude selection process: a gradient update is either salient, in which case it will be applied quickly, or is eventually rendered insignificant by the error accumulation process, in which case it need not have been applied in the first place. This intuition holds for both convex and non-convex objectives, although the technical details are different.

**Related Work.** There has been a recent surge of interest in distributed machine learning, e.g., [1, 33, 6]; due to space limits, we focus on communication-reduction techniques that are closely related.

**Lossless Methods.** One way of doing *lossless* communication-reduction is through *factorization* [7, 31], which is effective in deep neural networks with large fully-connected layers, whose gradients can be decomposed as outer vector products. This method is not generally applicable, and in particular may not be efficient in networks with large convolutional layers, e.g., [13, 27]. A second lossless method is executing *extremely large batches*, hiding communication cost behind increased computation [11, 32]. Although promising, these methods currently require careful per-instance

parameter tuning, and do not eliminate communication costs. Asynchronous methods, e.g., [20] can also be seen as a way of performing communication-reduction, by overlapping communication and computation, but are also known to require careful parameter tuning [34].

**Quantization.** Seide et al. [23] and Strom [25] were among the first to propose quantization to reduce the bandwidth costs of training deep networks. Their techniques employ a variant of error-accumulation. Alistarh et al. [4, 12] introduced a theoretically-justified stochastic quantization technique called Quantized SGD (QSGD), which trades off compression and convergence rate. This technique was significantly refined for the case of two-bit precision by [30]. Recent work [28] studies the problem of selecting a *sparse, low-variance* unbiased gradient estimator as a linear planning problem. This approach differs from the algorithms we analyze, as it ensures unbiasedness of the estimators in every iteration. By contrast, error accumulation inherently biases the applied updates.

**Sparsification.** Strom [25], Dryden et al. [9] and Aji and Heafield [2] considered *sparsifying* the gradient updates by only applying the top $K$ components, taken at at every node, in every iteration, for $K$ corresponding to $< 1\%$ of the dimension, and accumulating the error. Shokri [24] and Sun et al. [26] independently considered similar algorithms, but for privacy and regularization purposes, respectively. Lin et al. [17] performed an in-depth empirical exploration of this space in the context of training neural networks, showing that extremely high gradient sparsity can be supported by convolutional and recurrent networks, without loss of accuracy, under careful hyperparameter tuning.

**Analytic Techniques.** The first reference to approach the analysis of quantization techniques is Buckwild! [8], in the context of asynchronous training of generalized linear models. Our analysis in the case of convex SGD uses similar notions of convergence, and a similar general approach. The distinctions are: 1) the algorithm we analyze is different; 2) we do not assume the existence of a bound $\tau$ on the delay with which a component may be applied; 3) we do not make sparsity assumptions on the original stochastic gradients. In the non-convex case, we use a different approach.

## 2 Preliminaries

**Background and Assumptions.** Please recall our modeling of the basic SGD process in Equation (1). Fix $n$ to be the dimension of the problems we consider; unless otherwise stated $\|\cdot\|$ will denote the 2-norm. We begin by considering a general setting where SGD is used to minimize a function $f : \mathbb{R}^n \to \mathbb{R}$, which can be either convex or non-convex, using unbiased stochastic gradient samples $\tilde{G}(\cdot)$, i.e., $\mathbf{E}[\tilde{G}(x_t)] = \nabla f(x_t)$.

We assume throughout the paper that the second moment of the average of $P$ stochastic gradients with respect to any choice of parameter values is bounded, i.e.:

$$\mathbf{E}[\|\frac{1}{P}\sum_{p=1}^{P}\tilde{G}^p(x)\|^2] \leq M^2, \forall x \in \mathbb{R}^n \tag{2}$$

where $\tilde{G}^1(x), \ldots, \tilde{G}^P(x)$ are $P$ independent stochastic gradients (at each node). We also give the following definitions:

**Definition 1.** For any differentiable function $f: \mathbb{R}^d \to \mathbb{R}$,

- $f$ is $c$-strongly convex if $\forall x, y \in \mathbb{R}^d$, it satisfies $f(y) \geq f(x) + \langle \nabla f(x), y - x \rangle + \frac{c}{2}\|x - y\|^2$.
- $f$ is $L$-Lipschitz smooth (or $L$-smooth for short) if $\forall x, y \in \mathbb{R}^d$, $\|\nabla f(x) - \nabla f(y)\| \leq L\|x - y\|$.

We consider both $c$-strongly convex and $L$-Lipschitz smooth (non-convex) objectives. Let $x^*$ be the optimum parameter set minimizing Equation (1). For $\epsilon > 0$, the "success region" to which we want to converge is the set of parameters $S = \{x \mid \|x - x^*\|^2 \leq \epsilon\}$.

**Rate Supermartingales.** In the convex case, we phrase convergence of SGD in terms of rate supermartingales; we will follow the presentation of De et al. [8] for background. A *supermartingale* is a stochastic process $W_t$ with the property that that $\mathbf{E}[W_{t+1}|W_t] \leq W_t$. A martingale-based proof of convergence will construct a supermartingale $W_t(x_t, x_{t-1}, \ldots, x_0)$ that is a function of time and the current and previous iterates; it intuitively represents how far the algorithm is from convergence.

**Definition 2.** Given a stochastic algorithm such as the iteration in Equation (1), a non-negative process $W_t : \mathbb{R}^{n \times t} \to \mathbb{R}$ is a *rate supermartingale* with horizon $B$ if the following conditions are true. First, it must be a supermartingale: for any sequence $x_t, \ldots, x_0$ and any $t \leq B$,

$$\mathbf{E}[W_{t+1}(x_t - \alpha\tilde{G}_t(x_t), x_t, \ldots, x_0)] \leq W_t(x_t, x_{t-1}, \ldots, x_0). \tag{3}$$

---

**Algorithm 1** Parallel TopK SGD at a node $p$.

---

**Input:** Stochastic Gradient Oracle $\tilde{G}^p(\cdot)$ at node $p$
**Input:** value $K$, learning rate $\alpha$
Initialize $v_0 = \epsilon_0^p = \vec{0}$
**for** each step $t \geq 1$ **do**
    $acc_t^p \leftarrow \epsilon_{t-1}^p + \alpha \tilde{G}_t^p(v_{t-1})$ {accumulate error into a locally generated gradient}
    $\epsilon_t^p \leftarrow acc_t^p - \mathsf{TopK}(acc_t^p)$ {update the error}
    $\mathsf{Broadcast}(\mathsf{TopK}(acc_t^p), \mathsf{SUM})$ { broadcast to all nodes and receive from all nodes }
    $g_t \leftarrow \frac{1}{P} \sum_{q=1}^{P} \mathsf{TopK}(acc_t^q)$ { average the received (sparse) gradients }
    $v_t \leftarrow v_{t-1} - g_t$ { apply the update }
**end for**

---

Second, for all times $T \leq B$ and for any sequence $x_T, \ldots, x_0$, if the algorithm has not succeeded in entering the success region $S$ by time $T$, it must hold that

$$W_T(x_T, x_{T-1}, \ldots, x_0) \geq T. \tag{4}$$

**Convergence.** Assuming the existence of a rate supermartingale, one can bound the convergence rate of the corresponding stochastic process.

**Statement 1.** *Assume that we run a stochastic algorithm, for which $W$ is a* rate supermartingale. *For $T \leq B$, the probability that the algorithm does not complete by time $T$ is*

$$Pr(F_T) \leq \frac{\mathbf{E}[W_0(x_0)]}{T}.$$

The proof of this general fact is given by De Sa et al. [8], among others. A rate supermartingale for sequential SGD is:

**Statement 2** ([8]). *There exists a $W_t$ where, if the algorithm has not succeeded by timestep $t$,*

$$W_t(x_t, \ldots, x_0) = \frac{\epsilon}{2\alpha c \epsilon - \alpha^2 \tilde{M}^2} \log\left(e \left\| x_t - x^* \right\|^2 \epsilon^{-1}\right) + t,$$

*where $\tilde{M}$ is a bound on the second moment of the stochastic gradients for the sequential SGD process. Further, $W_t$ is a rate submartingale for sequential SGD with horizon $B = \infty$. It is also $H$-Lipschitz in the first coordinate, with $H = 2\sqrt{\epsilon}\left(2\alpha c \epsilon - \alpha^2 M^2\right)^{-1}$, that is for any $t, u, v$ and any sequence $x_{t-1}, \ldots, x_0 : \left\| W_t\left(u, x_{t-1}, \ldots, x_0\right) - W_t\left(v, x_{t-1}, \ldots, x_0\right) \right\| \leq H \left\| u - v \right\|.$*

## 3 The TopK SGD Algorithm

**Algorithm Description.** In the following, we will consider a variant of distributed SGD where, in each iteration $t$, each node computes a local gradient based on its current view of the model, which we denote by $v_t$, which is consistent across nodes (see Algorithm 1 for pseudocode). The node adds its local error vector from the previous iteration (defined below) into the gradient, and then *truncates* this sum to its top $K$ components, sorted in decreasing order of (absolute) magnitude. Each node accumulates the components which were not selected locally into the error vector $\epsilon_t$, which is added to the current gradient *before* the truncation procedure. The selected top $K$ components are then broadcast to all other nodes. (We assume that broadcast happens point-to-point, but in practice it could be intermediated by a parameter server, or via a more complex reduction procedure.) Each node collects all messages from its peers, and applies their average to the local model. This update is the same across all nodes, and therefore $v_t$ is consistent across nodes at every iteration.

Variants of this pattern are implemented in [2, 9, 17, 25, 26]. When training networks, this pattern is used in conjunction with heuristics such as momentum tuning and gradient clipping [17].

**Analysis Preliminaries.** Define $\tilde{G}_t(v_t) = \frac{1}{P} \sum_{p=1}^{P} \tilde{G}_t^p(v_t)$. In the following, it will be useful to track the following auxiliary random variable at each global step $t$:

$$x_{t+1} = x_t - \frac{1}{P} \sum_{p=1}^{P} \alpha \tilde{G}_t^p(v_t) = x_t - \alpha \tilde{G}_t(v_t), \tag{5}$$

where $x_0 = 0^n$. Intuitively, $x_t$ tracks all the gradients generated so far, without truncation. One of our first objectives will be to bound the difference between $x_t$ and $v_t$ at each time step $t$. Define:

$$\epsilon_t = \frac{1}{P} \sum_{p=1}^{P} \epsilon_t^p. \tag{6}$$

The variable $x_t$ is set up such that, by induction on $t$, one can prove that, for any time $t \geq 0$,

$$v_t - x_t = \epsilon_t. \tag{7}$$

**Convergence.** A reasonable question is whether we wish to show convergence with respect to the auxiliary variable $x_t$, which aggregates gradients, or with respect to the variable $v_t$, which measures convergence in the *view* which only accumulates truncated gradients. Our analysis will in fact show that the TopK algorithm converges in *both* these measures, albeit at slightly different rates. So, in particular, nodes will be able to observe convergence by directly observing the "shared" parameter $v_t$.

## 3.1 An Analytic Assumption

The update to the parameter $v_{t+1}$ at each step is

$$\frac{1}{P} \sum_{p=1}^{P} \mathsf{TopK}\left(\alpha \tilde{G}_t^p(v_t) + \epsilon_t^p\right).$$

The intention is to apply the top $K$ components of the sum of updates across all nodes, that is,

$$\frac{1}{P} \mathsf{TopK}\left(\sum_{p=1}^{P} \left(\alpha \tilde{G}_t^p(v_t) + \epsilon_t^p\right)\right).$$

However, it may well happen that these two terms are different: one could have a fixed component $j$ of $\alpha \tilde{G}_t^p + \epsilon_t^p$ with the large absolute values, but opposite signs, at two distinct nodes, and value 0 at all other nodes. This component would be selected at these two nodes (since it has high absolute value locally), whereas it would not be part of the top $K$ taken over the total sum, since its contribution to the sum would be close to 0. Obviously, if this were to happen on all components, the algorithm would make very little progress in such a step.

In the following, we will assume that such overlaps can only cause the algorithm to lose a small amount of information at each step, with respect to the norm of "true" gradient $\tilde{G}_t$. Specifically:

**Assumption 1.** *There exists a (small) constant $\xi$ such that, for every iteration $t \geq 0$, we have:*

$$\left\| \mathsf{TopK}\left(\frac{1}{P} \sum_{p=1}^{P} \left(\alpha \tilde{G}_t^p(v_t) + \epsilon_t^p\right)\right) - \sum_{p=1}^{P} \frac{1}{P} \mathsf{TopK}\left(\alpha \tilde{G}_t^p(v_t) + \epsilon_t^p\right) \right\| \leq \xi \|\alpha \tilde{G}_t(v_t)\|. \tag{8}$$

**Discussion.** We validate Assumption 1 experimentally on a number of different learning tasks in Section 6 (see also Figure 1). In addition, we emphasize the following points:

- As per our later analysis, in both the convex *and* non-convex cases, the influence of $\xi$ on convergence is dampened linearly by the number of nodes $P$. Unless $\xi$ grows linearly with $P$, which appears unlikely, its value will become irrelevant as parallelism is increased.
- Assumption 1 is necessary for a general, worst-case analysis. Its role is to bound the gap between the top-K of the gradient sum (which would be applied at each step in a "sequential" version of the process), and the sum of top-Ks (which is applied in the distributed version). If the number of nodes $P$ is 1, the assumption trivially holds.
  To illustrate necessity, consider a dummy instance with two nodes, dimension 2, and $K = 1$. Assume that at a step node 1 has gradient vector $(-1001, 500)$, and node 2 has gradient vector $(1001, 500)$. Selecting the top-1 ($\max$ abs) of the sum of the two gradients would result in the gradient $(0, 1000)$. Applying the sum of top-1's taken locally results in the gradient $(0, 0)$, since we select $(1001, 0)$ and $(-1001, 0)$, respectively. This is clearly not desirable, but in theory possible. The assumption states that this worst-case scenario is unlikely, by bounding the norm difference between the two terms.

- The intuitive cause for the example above is the high variability of the local gradients at the nodes. One can therefore view Assumption 1 as a bound on the variance of the local gradients (at the nodes) with respect to the global variance (aggregated over all nodes). We further expand on this observation in Section 6.

# 4 Analysis in the Convex Case

We now focus on the convergence of Algorithm 1 with respect to the parameter $v_t$. We assume that the function $f$ is $c$-strongly convex and that the bound (2) holds. Due to space constraints, the complete proofs are deferred to the full version of our paper [3].

**Technical Preliminaries.** We begin by noting that for any vector $x \in \mathbb{R}^n$, it holds that

$$\|x - \mathsf{TopK}\,(x)\,\|_1 \leq \frac{n-K}{n}\|x\|_1, \text{ and } \|x - \mathsf{TopK}\,(x)\,\|^2 \leq \frac{n-K}{n}\|x\|^2.$$

Thus, if $\gamma = \sqrt{\frac{n-K}{n}}$, we have that $\|x - \mathsf{TopK}\,(x)\,\| \leq \gamma\|x\|$. In practice, the last inequality may be satisfied by a much smaller value of $\gamma$, since the gradient values are very unlikely to be uniform. We now bound the difference between $v_t$ and $x_t$ using Assumption 1. We have the following:

**Lemma 1.** *With the processes $x_t$ and $v_t$ defined as above:*

$$
\begin{aligned}
\|v_t - x_t\| &= \left\| \frac{1}{P} \sum_{p=1}^{P} \left( \alpha \tilde{G}_{t-1}^p(v_{t-1}) + \epsilon_{t-1}^p \right) - \frac{1}{P} \sum_{p=1}^{P} \mathsf{TopK}\left( \alpha \tilde{G}_{t-1}^p(v_{t-1}) + \epsilon_{t-1}^p \right) \right\| \\
&\leq \left( \gamma + \frac{\xi}{P} \right) \sum_{k=1}^{t} \gamma^{k-1} \|x_{t-k+1} - x_{t-k}\|.
\end{aligned}
\tag{9}
$$

We now use the previous result to bound a quantity that represents the difference between the updates based on the TopK procedure and those based on full gradients.

**Lemma 2.** *Under the assumptions above, taking expectation with respect to gradients at time $t$:*

$$
\begin{aligned}
\mathbf{E} &\left[ \left\| \frac{1}{P} \sum_{p=1}^{P} \left( \alpha \tilde{G}_t^p(v_t) \right) - \frac{1}{P} \sum_{p=1}^{P} \mathsf{TopK}\left( \alpha \tilde{G}_t^p(v_t) + \epsilon_t^p \right) \right\| \right] \\
&\leq (\gamma + 1)\left( \gamma + \frac{\xi}{P} \right) \sum_{k=1}^{t} \gamma^{k-1} \|x_{t-k+1} - x_{t-k}\| + \left( \gamma + \frac{\xi}{P} \right) \alpha M.
\end{aligned}
\tag{10}
$$

Before we move on, we must introduce some notation. Set constants

$$C = (\gamma + 1)\left( \gamma + \frac{\xi}{P} \right) \sum_{k=1}^{\infty} \gamma^{k-1} = \frac{1+\gamma}{1-\gamma}\left( \gamma + \frac{\xi}{P} \right),$$

and

$$C' = C + \left( \gamma + \frac{\xi}{P} \right) = \left( \gamma + \frac{\xi}{P} \right) \frac{2}{1-\gamma}.$$

**The Convergence Bound.** Our main result in this section is the following:

**Theorem 1.** *Assume that $W$ is a rate supermartingale with horizon $B$ for the sequential SGD algorithm and that $W$ is $H$-Lipschitz in the first coordinate. Assume further that $\alpha H M C' < 1$. Then for any $T \leq B$, the probability that $v_s \notin S$ for all $s \leq T$ is:*

$$Pr\,[F_T] \leq \frac{\mathbf{E}\,[W_0\,(v_0)]}{(1 - \alpha H M C')\,T}.\tag{11}$$

The proof proceeds by defining a carefully-designed random process with respect to the iterate $v_t$, and proving that it is a rate supermartingale assuming the existence of $W$. We now apply this result with the martingale $W_t$ for the sequential SGD process that uses the average of $P$ stochastic gradients as an update (so that $\tilde{M} = M$ in Statement 2). We obtain:

**Corollary 1.** *Assume that we run Algorithm 1 for minimizing a convex function $f$ satisfying the listed assumptions. Suppose that the learning rate is set to $\alpha$, with:*

$$\alpha < \min \left\{ \frac{2c\epsilon}{M^2}, \frac{2\left(c\epsilon - \sqrt{\epsilon}MC'\right)}{M^2} \right\}.$$

*Then for any $T > 0$ the probability that $v_i \notin S$ for all $i \leq T$ is:*

$$Pr\left(F_T\right) \leq \frac{\epsilon}{\left(2\alpha c\epsilon - \alpha^2 M^2 - \alpha 2\sqrt{\epsilon}MC'\right)T} \log \left( \frac{e\|v_0 - x^*\|^2}{\epsilon} \right). \tag{12}$$

Note that the learning rate is chosen so that the denominator on the right-hand side is positive. This is discussed in further detail in Section 6. Compared to the sequential case (Statement 2), the convergence rate for the TopK algorithm features a slowdown of $\alpha 2\sqrt{\epsilon}MC'$. Assuming that $P$ is constant with respect to $n/K$,

$$C' = \left( \sqrt{\frac{n-K}{n}} + \frac{\xi}{P} \right) \frac{2}{1 - \sqrt{\frac{n-K}{n}}} = 2\frac{n}{K} \left( \sqrt{\frac{n-K}{n}} + \frac{\xi}{P} \right) \left( 1 + \sqrt{\frac{n-K}{n}} \right) = \mathcal{O}\left( \frac{n}{K} \right).$$

Hence, the slowdown is linear in $n/K$ and $\xi/P$. In particular, the effect of $\xi$ is dampened by the number of nodes.

## 5 Analysis for the Non-Convex Case

We now consider the more general case when SGD is minimizing a (not necessarily convex) function $f$, using SGD with (decreasing) step sizes $\alpha_t$. Again, we assume that the bound (2) holds. We also assume that $f$ is $L$-Lipschitz smooth.

As is standard in non-convex settings [18], we settle for a weaker notion of convergence, namely:

$$\min_{t\in\{1,\ldots,T\}} \mathbf{E}\left[\|\nabla f\left(v_t\right)\|^2\right] \overset{T\to\infty}{\longrightarrow} 0,$$

that is, the algorithm converges ergodically to a point where gradients are 0. Our strategy will be to leverage the bound on the difference between the "real" model $x_t$ and the view $v_t$ observed at iteration $t$ to bound the expected value of $f(v_t)$, which in turn will allow us to bound

$$\frac{1}{\sum_{t=1}^{T} \alpha_t} \sum_{t=1}^{T} \alpha_t \mathbf{E}\left[\|\nabla f\left(v_t\right)\|^2\right],$$

where the parameters $\alpha_t$ are appropriately chosen *decreasing* learning rate parameters. We start from:

**Lemma 3.** *For any time $t \geq 1$: $\|v_t - x_t\|^2 \leq \left(1 + \frac{\xi}{P\gamma}\right)^2 \sum_{k=1}^{t} \left(2\gamma^2\right)^k \|x_{t-k+1} - x_{t-k}\|^2$.*

We will leverage this bound on the gap to prove the following general bound:

**Theorem 2.** *Consider the TopK algorithm for minimising a function $f$ that satisfies the assumptions in this section. Suppose that the learning rate sequence and $K$ are chosen so that for any time $t > 0$:*

$$\sum_{k=1}^{t} \left(2\gamma^2\right)^k \frac{\alpha_{t-k}^2}{\alpha_t} \leq D \tag{13}$$

*for some constant $D > 0$. Then, after running Algorithm 1 for $T$ steps:*

$$\frac{1}{\sum_{t=1}^{T} \alpha_t} \sum_{t=1}^{T} \alpha_t \mathbf{E}\left[\|\nabla f\left(v_t\right)\|^2\right] \leq \frac{4\left(f\left(x_0\right) - f\left(x^*\right)\right)}{\sum_{t=1}^{T} \alpha_t}$$

$$+ \frac{\left(2LM^2 + 4L^2M^2\left(1 + \frac{\xi}{P\gamma}\right)^2 D\right)\sum_{t=1}^{T} \alpha_t^2}{\sum_{t=1}^{T} \alpha_t}. \tag{14}$$

Notice again that the effect of $\xi$ in the bound is dampened by $P$. One can show that inequality (13) holds whenever $K = cn$ for some constant $c > \frac{1}{2}$ and the step sizes are chosen so that $\alpha_t = t^{-\theta}$ for a constant $\theta > 0$. When $K = cn$ with $c > \frac{1}{2}$, a constant learning rate depending on the number of iterations $T$ can also be used to ensure ergodic convergence. We refer the reader to the full version of our paper for a complete derivation [3].

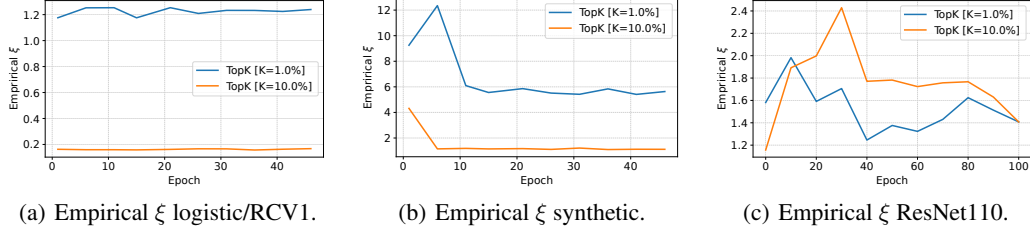

(a) Empirical $\xi$ logistic/RCV1.   (b) Empirical $\xi$ synthetic.   (c) Empirical $\xi$ ResNet110.

Figure 1: Validating Assumption 1 on various models and datasets.

## 6  Discussion and Experimental Validation

**The Analytic Assumption.** We start by empirically validating Assumption 1 in Figure 1 on two regression tasks (a synthetic linear regression task of dimension 1,024, and logistic regression for text categorization on RCV1 [15]), as well as ResNet110 [13] on CIFAR-10 [14]. Exact descriptions of the experimental setup are given in the full version of the paper [5]. Specifically, we sample gradients at different epochs during the training process, and bound the constant $\xi$ by comparing the left and right-hand sides of Equation (8). The assumption appears to hold with relatively low, stable values of the constant $\xi$. We note that RCV1 is relatively sparse (average density $\simeq 10\%$), while gradients in the other two settings are fully dense.

Additionally, we present an intuitive justification why Assumption 1 can be seen as a bound on the variance of the local gradients with respect to the global variance. Through a series of elementary operations, one can obtain:

$$\|\epsilon_t\| \le \frac{1}{P}\sum_{p=1}^{P}\|\epsilon_t^p\| \le \frac{1}{P}\sum_{k=1}^{t}\gamma^k\sum_{p=1}^{P}\|\alpha\tilde{G}_{t-k+1}^p\|,\tag{15}$$

which in turn implies that:

$$\|\mathsf{TopK}\left(\frac{1}{P}\sum_{p=1}^{P}\left(\alpha\tilde{G}_t^p(v_t)+\epsilon_t^p\right)\right) - \sum_{p=1}^{P}\frac{1}{P}\mathsf{TopK}\left(\alpha\tilde{G}_t^p(v_t)+\epsilon_t^p\right)\|$$

$$\le \gamma\alpha\|\tilde{G}_t\| + \frac{\gamma}{P}\|\epsilon_t\| + \frac{\gamma}{P}\sum_{p=1}^{P}\|\epsilon_t^p\| + \frac{\gamma\alpha}{P}\sum_{p=1}^{P}\|\alpha\tilde{G}_t^p(v_t)\|\tag{16}$$

The left-hand side of (16) is the quantity we wanted to control via Assumption 1. The first term on the right-hand side is the global (averaged) gradient at time $t$, while the remaining terms are all bounded by a dampened sum of local gradients, as per equation (15). Therefore, assuming a bound on the variance of the local gradients with respect to the global variance is equivalent to saying that the left-hand side of (16) is bounded by the the norm of the global gradient, at least in expectation. This is exactly the intention behind Assumption 1.

Note also that equation (15) provides a bound on the norm of the error term at time $t$, which is similar to the one in Lemma 1, but expressed in terms of the norms of the *local gradients*. One can build on this argument and our techniques in Section 4 to show convergence of the TopK algorithm directly. However, such analysis will rely on a bound on the variance of the *local gradients* (as apposed to the bound in equation (2)), which is a strong assumption that ignores the effect of averaging over the $P$ nodes. In contrast, Assumption 1 allows for a more elegant analysis that provides better convergence rates, which are due to the averaging of the local gradients at every step of the TopK algorithm. We refer to the full version of our paper for further details.

**Learning Rate and Variance.** In the convex case, the choice of learning rate must ensure both

$$2\alpha c\epsilon - \alpha^2 M^2 > 0 \text{ and } \alpha HMC' < 1, \text{ implying } \alpha < \min\left\{\frac{2c\epsilon}{M^2}, \frac{2\left(c\epsilon - \sqrt{\epsilon}MC'\right)}{M^2}\right\}.\tag{17}$$

Note that this requires the second term to be positive, that is $\epsilon > \left(\frac{MC'}{c}\right)^2$. Hence, if we aim for convergence within a small region around the optimum, we may need to ensure that gradient variance is bounded, either by minibatching or, empirically, by gradient clipping [17].

**The Impact of the Parameter $K$ and Gradient "Shape."** In the convex case, the dependence on the convergence with respect to $K$ and $n$ is encapsulated by the parameter $C' = \mathcal{O}(n/K)$ assuming $P$ is constant. Throughout the analysis, we only used worst-case bounds on the norm gap between the gradient and its top $K$ components. These bounds are tight in the (unlikely) case where the gradient values are uniformly distributed; however, there is empirical evidence showing that this is not the case in practice [19], suggesting that this gap should be smaller. The algorithm may implicitly exploit this narrower gap for improved convergence. Please see Figure 2 for empirical validation of this claim, confirming that the gradient norm is concentrated towards the top elements.

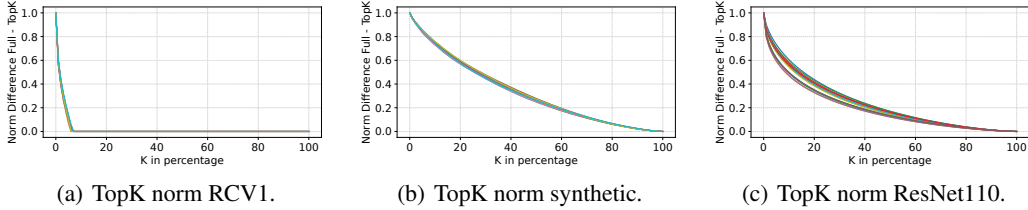

(a) TopK norm RCV1.     (b) TopK norm synthetic.     (c) TopK norm ResNet110.

Figure 2: Examining the value of $\|\tilde{G} - \mathsf{TopK}(\tilde{G})\|/\|\tilde{G}\|$ versus $K$ on various datasets/tasks. Every line represents a randomly chosen gradient per epoch during training with standard hyper parameters.

In the non-convex case, the condition $K = cn$ with $c > 1/2$ is quite restrictive. Again, the condition is required since we are assuming the worst-case configuration (uniform values) for the gradients, in which case the bound in Lemma 4 is tight. However, we argue that in practice gradients are unlikely to be uniformly distributed; in fact, empirical studies [19] have noticed that usually gradient components are normally distributed, which should enable us to improve this lower bound on $c$.

**Comparison with SGD Variants.** In the convex case, we note that, when $K$ is a constant fraction of $n$, the convergence of the TopK algorithm is essentially dictated by the Lipschitz constant of the supermartingale $W$, and by the second-moment bound $M$, and will be similar to sequential SGD. Please see Figure 3 for an empirical validation of this fact.

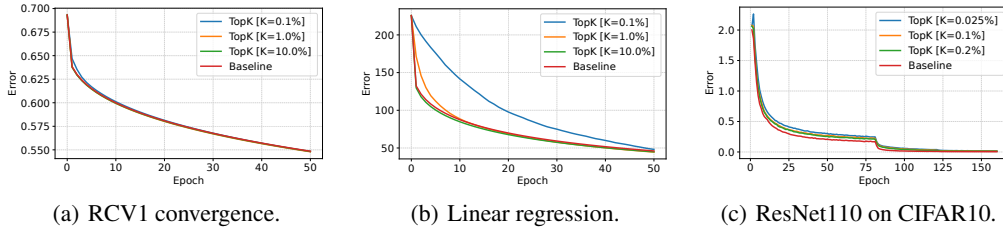

(a) RCV1 convergence.     (b) Linear regression.     (c) ResNet110 on CIFAR10.

Figure 3: Examining convergence versus value of $K$ on various datasets and tasks.

Compared to asynchronous SGD, the convergence rate of the TopK algorithm is basically that of an asynchronous algorithm with maximum delay $\tau = \mathcal{O}(\sqrt{n}/K)$. That is because an asynchronous algorithm with dense updates and max delay $\tau$ has a convergence slowdown of $\Theta(\tau\sqrt{n})$ [8, 16, 3]. We note that, for large sparsity ($0.1\%$—$1\%$), there is a noticeable convergence slowdown, as predicted. The worst-case convergence of TopK is similar to SGD with stochastic quantization, e.g., [4, 28]: for instance, for $K = \sqrt{n}$, the worst-case convergence slowdown is $\mathcal{O}(\sqrt{n})$, the same as QSGD [4]. The TopK procedure is arguably simpler to implement than the parametrized quantization and encoding techniques required to make stochastic quantization behave well [4]. Here, TopK had superior convergence rate compared to stochastic quantization/sparsification [4, 28] given the same communication budget per node.

## 7 Conclusions

We provided the first theoretical analysis of the "TopK" sparsification communication-reduction technique. Our approach should extend to methods combining sparsification with quantization by reduced precision [2, 25] and methods using approximate quantiles [2, 17]. We provide a theoretical foundation for empirical results shown with large-scale experiments on recurrent neural networks on production-scale speech, neural machine translation, as well as image classification tasks [9, 17, 25, 2].

## Acknowledgement

This project has received funding from the European Union's Horizon 2020 research and innovation programme under the Marie Skłodowska-Curie Grant Agreement No. 665385.

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
