[Supplementary Material]

# A   Analysis for the Convex Case

**Lemma 1.** *With the processes $x_t$ and $v_t$ defined as above:*

$$\|v_t - x_t\| = \|\frac{1}{P}\sum_{p=1}^{P}\left(\alpha\tilde{G}_{t-1}^p(v_{t-1}) + \epsilon_{t-1}^p\right) - \frac{1}{P}\sum_{p=1}^{P}\mathsf{TopK}\left(\alpha\tilde{G}_{t-1}^p(v_{t-1}) + \epsilon_{t-1}^p\right)\|$$

$$\leq \left(\gamma + \frac{\xi}{P}\right)\sum_{k=1}^{t}\gamma^{k-1}\|x_{t-k+1} - x_{t-k}\|.$$

(18)

*Proof.* First, we obtain a recursive relation of the form:

$$\|v_{t+1} - x_{t+1}\| = \left\|v_t - x_t + \frac{1}{P}\sum_{p=1}^{P}\left(\alpha\tilde{G}_t^p(v_t) + \epsilon_t^p\right) - \epsilon_t - \frac{1}{P}\sum_{p=1}^{P}\mathsf{TopK}\left(\alpha\tilde{G}_t^p(v_t) + \epsilon_t^p\right)\right\|$$

$$\overset{(7)}{=}\left\|\frac{1}{P}\sum_{p=1}^{P}\left(\alpha\tilde{G}_t^p(v_t) + \epsilon_t^p\right) - \frac{1}{P}\sum_{p=1}^{P}\mathsf{TopK}\left(\alpha\tilde{G}_t^p(v_t) + \epsilon_t^p\right)\right\|$$

$$=\|\frac{1}{P}\sum_{p=1}^{P}\left(\alpha\tilde{G}_t^p(v_t) + \epsilon_t^p\right) - \frac{1}{P}\mathsf{TopK}\left(\sum_{p=1}^{P}\left(\alpha\tilde{G}_t^p(v_t) + \epsilon_t^p\right)\right) +$$

$$+\frac{1}{P}\mathsf{TopK}\left(\sum_{p=1}^{P}\left(\alpha\tilde{G}_t^p(v_t) + \epsilon_t^p\right)\right) - \frac{1}{P}\sum_{p=1}^{P}\mathsf{TopK}\left(\alpha\tilde{G}_t^p(v_t) + \epsilon_t^p\right)\|$$

$$\leq\frac{\gamma}{P}\|\sum_{p=1}^{P}\left(\alpha\tilde{G}_t^p(v_t) + \epsilon_t^p\right)\| + \frac{\xi}{P}\|\alpha\tilde{G}_t(v_t)\|$$

$$=\gamma\|\alpha\tilde{G}_t(v_t) + v_t - x_t\| + \frac{\xi}{P}\|\alpha\tilde{G}_t(v_t)\|$$

$$\leq\gamma\|v_t - x_t\| + \left(\gamma + \frac{\xi}{P}\right)\|x_{t+1} - x_t\|$$

Iterating this downwards yields the result. □

Next, we use the previous result to bound a quantity that represents the difference between the updates based on the TopK procedure and those based on full gradients.

**Lemma 2.** *Under the assumptions above and with expectation taken with respect to the gradients at time $t$:*

$$\mathbf{E}\left[\|\frac{1}{P}\sum_{p=1}^{P}\left(\alpha\tilde{G}_t^p(v_t)\right) - \frac{1}{P}\sum_{p=1}^{P}\mathsf{TopK}\left(\alpha\tilde{G}_t^p(v_t) + \epsilon_t^p\right)\|\right] \leq (\gamma+1)\left(\gamma + \frac{\xi}{P}\right)\sum_{k=1}^{t}\gamma^{k-1}\|x_{t-k+1} - x_{t-k}\|$$

$$+ \left(\gamma + \frac{\xi}{P}\right)\alpha M$$

(19)

*Proof.* Using the result from Lemma 1:

$$\mathbf{E}\left[\|\frac{1}{P}\sum_{p=1}^{P}\left(\alpha\tilde{G}_t^p(v_t)\right) - \frac{1}{P}\sum_{p=1}^{P}\mathsf{TopK}\left(\alpha\tilde{G}_t^p(v_t)+\epsilon_t^p\right)\|\right]$$

$$\leq \mathbf{E}\left[\|\epsilon_t\|\right] + \mathbf{E}\left[\|\frac{1}{P}\sum_{p=1}^{P}\left(\alpha\tilde{G}_t^p(v_t)+\epsilon_t^p\right) - \frac{1}{P}\sum_{p=1}^{P}\mathsf{TopK}\left(\alpha\tilde{G}_t^p(v_t)+\epsilon_t^p\right)\|\right]$$

$$\leq \|\epsilon_t\| + \gamma\|v_t - x_t\| + \left(\gamma + \frac{\xi}{P}\right)\mathbf{E}\left[\|\alpha\tilde{G}(v_t)\|\right]$$

$$\leq (\gamma+1)\|v_t - x_t\| + \left(\gamma + \frac{\xi}{P}\right)\alpha M$$

$$\leq (\gamma+1)\left(\gamma + \frac{\xi}{P}\right)\sum_{k=1}^{t}\gamma^{k-1}\|x_{t-k+1} - x_{t-k}\| + \left(\gamma + \frac{\xi}{P}\right)\alpha M.$$

$\square$

Finally, we introduce some notation. Set

$$C = (\gamma+1)\left(\gamma + \frac{\xi}{P}\right)\sum_{k=1}^{\infty}\gamma^{k-1} = \frac{1+\gamma}{1-\gamma}\left(\gamma + \frac{\xi}{P}\right),$$

and

$$C' = C + \left(\gamma + \frac{\xi}{P}\right) = \left(\gamma + \frac{\xi}{P}\right)\frac{2}{1-\gamma}.$$

Note that

$$C' = \left(\sqrt{\frac{n-K}{n}} + \frac{\xi}{P}\right)\frac{2}{1 - \sqrt{\frac{n-K}{n}}} = 2\frac{n}{K}\left(\sqrt{\frac{n-K}{n}} + \frac{\xi}{P}\right)\left(1 + \sqrt{\frac{n-K}{n}}\right) = \mathcal{O}\left(\frac{n}{K}\right)$$

### A.1 The Main Result

We have the following:

**Theorem 1.** *Assume that $W$ is a rate supermartingale with horizon $B$ for the sequential SGD algorithm and that $W$ is $H$-Lipschitz in the first coordinate. Assume further that $\alpha HMC' < 1$. Then for any $T \leq B$, the probability that $v_s \notin S$ for all $s \leq T$ is:*

$$Pr\left[F_T\right] \leq \frac{\mathbf{E}\left[W_0\left(v_0\right)\right]}{(1-\alpha HMC')T}. \tag{20}$$

*Proof.* Consider the process, defined by:

$$V_t(v_t, \ldots, v_0) = W_t(v_t, \ldots, v_0) - \alpha HMCt + H\left((\gamma+1)\left(\gamma + \frac{\xi}{P}\right)\sum_{k=1}^{t}\|x_{t-k+1} - x_{t-k}\|\sum_{m=k}^{\infty}\gamma^{m-1}\right.$$
$$\left. - \left(\gamma + \frac{\xi}{P}\right)\alpha Mt\right),$$

if the algorithm has not succeeded by time $t$ (i.e. $x_s \notin S$ for all $s \leq T$) and by $V_t = V_{u-1}$ otherwise, where $u$ is the minimal index, such that $x_u \in S$.

In the case when the algorithm has not succeeded at time $t$, using $W$'s Lipschitz property:

$$V_{t+1}\left(v_{t+1}, v_t, \ldots, v_0\right) = W_{t+1}\left(v_t - \frac{1}{P}\sum_{p=1}^{P}\text{TopK}\left(\epsilon_t^p + \alpha\tilde{G}^p\left(v_t\right)\right), v_t, \ldots, v_0\right) - \alpha HMC\left(t+1\right)$$

$$+ H\left((\gamma+1)\left(\gamma + \frac{\xi}{P}\right)\sum_{k=1}^{t+1}\|x_{t-k+2} - x_{t-k+1}\|\sum_{m=k}^{\infty}\gamma^{m-1} - \left(\gamma + \frac{\xi}{P}\right)\alpha M\left(t+1\right)\right)$$

$$\leq W_{t+1}\left(v_t - \frac{1}{P}\sum_{p=1}^{P}\alpha\tilde{G}^p\left(v_t\right), v_t, \ldots, v_0\right)$$

$$+ H\|\frac{1}{P}\sum_{p=1}^{P}\alpha\tilde{G}^p\left(v_t\right) - \frac{1}{P}\sum_{p=1}^{P}\text{TopK}\left(\epsilon_t^p + \alpha\tilde{G}^p\left(v_t\right)\right)\|$$

$$- \alpha HMC\left(t+1\right) + H\left(1+\gamma\right)\left(\gamma + \frac{\xi}{P}\right)\|x_{t+1} - x_t\|\sum_{m=1}^{\infty}\gamma^{m-1}$$

$$+ H\left(\left(1+\gamma\right)\left(\gamma + \frac{\xi}{P}\right)\sum_{k=1}^{t}\|x_{t-k+1} - x_{t-k}\|\sum_{m=k+1}^{\infty}\gamma^{m-1} - \left(\gamma + \frac{\xi}{P}\right)\alpha M\left(t+1\right)\right)$$

Now we take expectation with respect to the randomness at time $t$ and conditional on the past. Note that the average of i.i.d. stochastic gradients is also a stochastic gradient. Using the supermartingale property of $W$, the bound on the expected norm of the gradient and (19):

$$\mathbf{E}\left[V_{t+1}\right] \leq W_t\left(v_t, \ldots, v_0\right) - \alpha HMCt + H\left((1+\gamma)\left(\gamma + \frac{\xi}{P}\right)\sum_{k=1}^{t}\|x_{t-k+1} - x_{t-k}\|\sum_{m=k}^{\infty}\gamma^{m-1}\right.$$

$$\left. - \left(\gamma + \frac{\xi}{P}\right)\alpha Mt\right) + \left(H\mathbf{E}\left[\|\alpha\tilde{G}\left(v_t\right)\|\right](1+\gamma)\left(\gamma + \frac{\xi}{P}\right)\sum_{m=1}^{\infty}\gamma^{m-1} - \alpha HMC\right)$$

$$+ H\left(\mathbf{E}\left[\|\frac{1}{P}\sum_{p=1}^{P}\alpha\tilde{G}^p\left(v_t\right) - \frac{1}{P}\sum_{p=1}^{P}\text{TopK}\left(\epsilon_t^p + \alpha\tilde{G}^p\left(v_t\right)\right)\|\right]\right.$$

$$\left. - (1+\gamma)\left(\gamma + \frac{\xi}{P}\right)\sum_{k=1}^{t}\|x_{t-k+1} - x_{t-k}\|\gamma^{k-1} - \left(\gamma + \frac{\xi}{P}\right)\alpha M\right)$$

$$\leq V_t.$$

The inequality also holds trivially in the case when the algorithm has succeeded at time $t$. Hence, $V_t$ is a supermartingale for the TopK process.

Now if the algorithm has not succeeded at time $T$, $W_T \geq T$, so $V_T \geq W_T - \alpha HMC'T \geq 0$. It follows that $V_T \geq 0$ for all $T$. Therefore,

$$\mathbf{E}\left[W_0\left(v_0\right)\right] = \mathbf{E}\left[V_0\left(v_0\right)\right]$$

$$\geq \mathbf{E}\left[V_T\right]$$

$$= \mathbf{E}\left[V_T | F_T\right]\Pr\left[F_T\right] + \mathbf{E}\left[V_T | \neg F_T\right]\Pr\left[\neg F_T\right]$$

$$\geq \mathbf{E}\left[V_T | F_T\right]\Pr\left[F_T\right]$$

$$= \mathbf{E}\left[W_T\left(v_T, \ldots, v_0\right) - \alpha HMCT + H\left((1+\gamma)\left(\gamma + \frac{\xi}{P}\right)\sum_{k=1}^{T}\|x_{T-k+1} - x_{T-k}\|\sum_{m=k}^{\infty}\gamma^{m-1}\right.\right.$$

$$\left.\left. - \left(\gamma + \frac{\xi}{P}\right)\alpha MT\right) | F_T\right]\Pr\left[F_T\right]$$

$$\geq \left(\mathbf{E}\left[W_T(v_T, \ldots, v_0) | F_T\right] - \alpha HM\left(C + \left(\gamma + \frac{\xi}{P}\right)\right)T\right)\Pr\left[F_T\right]$$

$$\geq \left(T - \alpha HMC'T\right)\Pr\left[F_T\right],$$

where we have used the fact that $W$ is a rate supermartingale. Hence we obtain:

$$\Pr\left[F_T\right] \le \frac{\mathbf{E}\left[W_0\left(x_0\right)\right]}{\left(1 - \alpha H M C'\right) T}.$$

$\square$

We now apply this result with a specific supermartingale $W$ for the sequential SGD process. Note that $W$ must be a supermartingale for the process that applies an average of $P$ updates, multiplied by the learning rate $\alpha$.
We use the following result from [8]:

**Lemma 3** ([8]). *Define the piecewise logarithm function to be*

$$\log(x) = \begin{cases} \log(ex) & : x \ge 1 \\ x & : x \le 1 \end{cases}$$

*Define the process $W_t$ by:*

$$W_t(x_t, \ldots, x_0) = \frac{\epsilon}{2\alpha c\epsilon - \alpha^2 M^2} \log\left(\|x_t - x^*\|^2 \epsilon^{-1}\right) + t,$$

*if the algorithm has not succeeded by timestep $t$ (i.e. $x_i \notin S$ for all $i \le t$) and by $W_t = W_{u-1}$ whenever $x_i \in S$ for some $i \le t$ and $u$ is the minimal index with this property. Then $W_t$ is a rate supermartingale for sequential SGD with horizon $B = \infty$. It is also $H$-Lipschitz in the first coordinate, with $H = 2\sqrt{\epsilon}\left(2\alpha c\epsilon - \alpha^2 M^2\right)^{-1}$, that is for any $t, u, v$ and any sequence $x_{t-1}, \ldots, x_0$:*

$$\|W_t\left(u, x_{t-1}, \ldots, x_0\right) - W_t\left(v, x_{t-1}, \ldots, x_0\right)\| \le H\|u - v\|.$$

Applying this particular martingale, we obtain:

**Corollary 1.** *Assume that we run Algorithm 1 for minimizing a convex function $f$ satisfying the listed assumptions. Suppose that the learning rate is set to $\alpha$, with:*

$$\alpha < \min\left\{\frac{2c\epsilon}{M^2}, \frac{2\left(c\epsilon - \sqrt{\epsilon}MC'\right)}{M^2}\right\}$$

*Then for any $T > 0$ the probability that $v_i \notin S$ for all $i \le T$ is:*

$$\mathbb{P}\left(F_T\right) \le \frac{\epsilon}{\left(2\alpha c\epsilon - \alpha^2 M^2 - \alpha 2\sqrt{\epsilon}MC'\right) T} \log\left(\frac{e\|v_0 - x^*\|^2}{\epsilon}\right). \tag{21}$$

*Proof.* Substituting and using the result from [8] that

$$\mathbb{E}\left(W_0\left(v_0\right)\right) \le \frac{\epsilon}{2\alpha c\epsilon - \alpha^2 M^2} \log\left(\frac{e\|v_0 - x^*\|^2}{\epsilon}\right)$$

we obtain that:

$$\mathbb{P}\left(F_T\right) \le \frac{\mathbb{E}\left(W_0\right)}{\left(1 - \alpha H M C'\right) T}$$

$$\le \frac{\epsilon}{2\alpha c\epsilon - \alpha^2 M^2} \log\left(\frac{e\|v_0 - x^*\|^2}{\epsilon}\right) \left(\left(1 - \alpha \frac{2\sqrt{\epsilon}}{2\alpha c\epsilon - \alpha^2 M^2} MC'\right) T\right)^{-1}$$

$$\le \frac{\epsilon}{\left(2\alpha c\epsilon - \alpha^2 M^2 - \alpha 2\sqrt{\epsilon}MC'\right) T} \log\left(\frac{e\|v_0 - x^*\|^2}{\epsilon}\right)$$

$\square$

# B   Analysis for the Non-Convex Case

**Setup.** We now consider the more general case when SGD is minimizing a (not necessarily convex) function $f$, using SGD with (decreasing) step sizes $\alpha_t$. Again, we assume that the second moment of the stochastic gradients is bounded in expectation (inequality (2)). Assume also that $\nabla f$ is $L$-Lipschitz (not only in expectation); that is, for all $x, y$:

$$\|\nabla f(x) - \nabla f(y)\| \leq L\|x - y\|. \tag{22}$$

As is standard in non-convex settings [17], will settle for a weaker notion of convergence, namely showing that

$$\min_{t \in \{1,\dots,T\}} \mathbf{E}\left[\|\nabla f(v_t)\|^2\right] \overset{T \to \infty}{\longrightarrow} 0,$$

that is, the algorithm converges ergodically to a local minimum of the function $f$. Our strategy will be to leverage our ability to bound the difference between the "real" model $x_t$ and the view $v_t$ observed at iteration $t$ to bound the evolution of the expected value of $f(v_t)$, which in turn will allow us to bound the sum

$$\frac{1}{\sum_{t=1}^{T} \alpha_k} \sum_{t=1}^{T} \alpha_t \mathbf{E}\left[\|\nabla f(v_t)\|^2\right],$$

where the parameters $\alpha_t$ are appropriately chosen *decreasing* learning rate parameters. This will enable us to show that the norm of the gradients converges towards zero in expectation.

We have the following:

**Lemma 4.** *For any time $t \geq 1$:*

$$\|v_t - x_t\|^2 \leq \left(1 + \frac{\xi}{P\gamma}\right)^2 \sum_{k=1}^{t} \left(2\gamma^2\right)^k \|x_{t-k+1} - x_{t-k}\|^2 \tag{23}$$

*Proof.* We had:

$$\|v_{t+1} - x_{t+1}\| \leq \gamma\|v_t - x_t\| + \left(\gamma + \frac{\xi}{P}\right)\|x_{t+1} - x_t\|$$

Hence,

$$\|v_{t+1} - x_{t+1}\|^2 \leq \left(\gamma\|v_t - x_t\| + \left(\gamma + \frac{\xi}{P}\right)\|x_{t+1} - x_t\|\right)^2 \leq 2\gamma^2\|v_t - x_t\|^2 + 2\left(\gamma + \frac{\xi}{P}\right)^2\|x_{t+1} - x_t\|^2$$

Iterating this gives:

$$\|v_t - x_t\|^2 \leq 2\left(\gamma + \frac{\xi}{P}\right)^2 \sum_{k=1}^{t} \left(2\gamma^2\right)^{k-1} \|x_{t-k+1} - x_{t-k}\|^2$$

$$= \left(1 + \frac{\xi}{P\gamma}\right)^2 \sum_{k=1}^{t} \left(2\gamma^2\right)^k \|x_{t-k+1} - x_{t-k}\|^2$$

$\square$

**Theorem 2.** *Consider the TopK algorithm for minimising a function $f$ that satisfies the above assumptions. Suppose that the learning rate sequence and $K$ are chosen so that for any time $t > 0$:*

$$\sum_{k=1}^{t} \left(2\gamma^2\right)^k \frac{\alpha_{t-k}^2}{\alpha_t} \leq D \tag{24}$$

*for some constant $D > 0$. Then, after running Algorithm 1 for $T$ steps:*

$$\frac{1}{\sum_{t=1}^{T} \alpha_t} \sum_{t=1}^{T} \alpha_t \mathbf{E}\left[\|\nabla f(v_t)\|^2\right] \leq \frac{4\left(f(x_0) - f(x^*)\right)}{\sum_{t=1}^{T} \alpha_t}$$

$$+ \frac{\left(2LM^2 + 4L^2M^2\left(1 + \frac{\xi}{P\gamma}\right)^2 D\right)\sum_{t=1}^{T} \alpha_t^2}{\sum_{t=1}^{T} \alpha_t} \tag{25}$$

*Proof of Theorem 2.* We begin by bounding the difference between the consecutive steps of the algorithm. By the assumption that $f$ is Lipschitz, for any time $t$:

$$f\left(x_{t+1}\right) - f\left(x_t\right) \leq \langle \nabla f\left(x_t\right), x_{t+1} - x_t \rangle + \frac{L}{2}\|x_{t+1} - x_t\|^2$$

$$= -\langle \nabla f\left(x_t\right), \alpha_t \tilde{G}_t\left(v_t\right) \rangle + \frac{L}{2}\|\alpha_t \tilde{G}_t\left(v_t\right)\|^2$$

Taking expectation with respect to the randomness at time $t$ and conditional on the past (denoted by $\mathbf{E}_{t|.}$):

$$\mathbf{E}_{t|.}\left[f\left(x_{t+1}\right)\right] - f\left(x_t\right) \leq -\alpha_t \langle \nabla f\left(x_t\right), \nabla f\left(v_t\right) \rangle + \frac{L}{2}\alpha_t^2 \mathbf{E}_{t|.}\left[\|\tilde{G}_t\left(v_t\right)\|^2\right]$$

$$= -\frac{\alpha_t}{2}\left(\|\nabla f\left(x_t\right)\|^2 + \|\nabla f\left(v_t\right)\|^2 - \|\nabla f\left(x_t\right) - \nabla f\left(v_t\right)\|^2\right)$$

$$+ \frac{L}{2}\alpha_t^2 \mathbf{E}_{t|.}\left[\|\tilde{G}_t\left(v_t\right)\|^2\right]$$

$$= -\frac{\alpha_t}{2}\|\nabla f\left(x_t\right)\|^2 - \frac{\alpha_t}{2}\|\nabla f\left(v_t\right)\|^2 + \frac{\alpha_t}{2}\|\nabla f\left(x_t\right) - \nabla f\left(v_t\right)\|^2$$

$$+ \frac{L}{2}\alpha_t^2 \mathbf{E}_{t|.}\left[\|\tilde{G}_t\left(v_t\right)\|^2\right]$$

$$\leq -\frac{\alpha_t}{2}\|\nabla f\left(x_t\right)\|^2 + \frac{\alpha_t}{2}L^2\|v_t - x_t\|^2 + \frac{L}{2}\alpha_t^2 \mathbf{E}_{t|.}\left[\|\tilde{G}_t\left(v_t\right)\|^2\right]$$

$$\leq -\frac{\alpha_t}{2}\left(\|\nabla f\left(x_t\right)\|^2 + L^2\|v_t - x_t\|^2\right) + \frac{L}{2}\alpha_t^2 M^2 + \alpha_t L^2\|v_t - x_t\|^2$$

Taking expectation with respect to the remaining gradients (before time $t$):

$$\mathbf{E}\left[f\left(x_{t+1}\right)\right] - \mathbf{E}\left[f\left(x_t\right)\right] \leq -\frac{\alpha_t}{2}\mathbf{E}\left[\|\nabla f\left(x_t\right)\|^2 + L^2\|v_t - x_t\|^2\right] + \frac{L}{2}\alpha_t^2 M^2 + \alpha_t L^2 \mathbf{E}\left[\|v_t - x_t\|^2\right]$$

$$(26)$$

But, using Lemma 4:

$$\mathbf{E}\left[\|v_t - x_t\|^2\right] \leq \left(1 + \frac{\xi}{P\gamma}\right)^2 \sum_{k=1}^{t}\left(2\gamma^2\right)^k \mathbf{E}\left[\|x_{t-k+1} - x_{t-k}\|^2\right]$$

$$\leq M^2\left(1 + \frac{\xi}{P\gamma}\right)^2 \alpha_t \sum_{k=1}^{t}\left(2\gamma^2\right)^k \frac{\alpha_{t-k}^2}{\alpha_t}$$

Now since for all $t$:

$$\sum_{k=1}^{t}\left(2\gamma^2\right)^k \frac{\alpha_{t-k}^2}{\alpha_t} \leq D$$

for some constant $D$, we have that:

$$\mathbf{E}\left[\|v_t - x_t\|^2\right] \leq M^2\left(1 + \frac{\xi}{P\gamma}\right)^2 \alpha_t D.$$

Therefore, we obtain:

$$\mathbf{E}\left[f\left(x_{t+1}\right)\right] - \mathbf{E}\left[f\left(x_t\right)\right] \leq -\frac{\alpha_t}{2}\mathbf{E}\left[\|\nabla f\left(x_t\right)\|^2 + L^2\|v_t - x_t\|^2\right] + \frac{L}{2}\alpha_t^2 M^2 + L^2 M^2\left(1 + \frac{\xi}{P\gamma}\right)^2 \alpha_t^2 D$$

Rearranging gives:

$$\alpha_t \mathbf{E}\left[\|\nabla f\left(x_t\right)\|^2 + L^2\|v_t - x_t\|^2\right] \leq 2\left(\mathbf{E}\left[f\left(x_t\right)\right] - \mathbf{E}\left[f\left(x_{t+1}\right)\right]\right)$$

$$+ \left(LM^2 + 2L^2 M^2\left(1 + \frac{\xi}{P\gamma}\right)^2 D\right)\alpha_t^2 \quad (27)$$

Note that, since the gradient is Lipschitz:

$$\|\nabla f\left(v_t\right)\|^2 = \|\left(\nabla f\left(v_t\right) - \nabla f\left(x_t\right)\right) + \nabla f\left(x_t\right)\|^2 \leq 2\|\nabla f\left(v_t\right) - \nabla f\left(x_t\right)\|^2 + 2\|\nabla f\left(x_t\right)\|^2$$

$$\leq 2L^2\|v_t - x_t\|^2 + 2\|\nabla f\left(x_t\right)\|^2$$

Applying this to the left-hand side of (27):

$$\alpha_t \mathbf{E}\left[\|\nabla f(v_t)\|^2\right] \le 4\left(\mathbf{E}\left[f(x_t)\right] - \mathbf{E}\left[f(x_{t+1})\right]\right)$$
$$+ \left(2LM^2 + 4L^2M^2\left(1 + \frac{\xi}{P\gamma}\right)^2 D\right)\alpha_t^2 \tag{28}$$

Now for any time $T$, summing over the bound in (28) and dividing by the sum of the learning rates:

$$\frac{1}{\sum_{t=1}^{T}\alpha_t}\sum_{t=1}^{T}\alpha_t\mathbf{E}\left[\|\nabla f(v_t)\|^2\right] \le \frac{4\left(f(x_0) - f(x^*)\right)}{\sum_{t=1}^{T}\alpha_t}$$
$$+ \frac{\left(2LM^2 + 4L^2M^2\left(1 + \frac{\xi}{P\gamma}\right)^2 D\right)\sum_{t=1}^{T}\alpha_t^2}{\sum_{t=1}^{T}\alpha_t} \tag{29}$$

$\square$

Therefore, it suffices to choose the learning rate sequence so that the term $\sum_{t=1}^{T}\alpha_t$ dominates $\sum_{t=1}^{T}\alpha_t^2$ asymptotically and so that the condition (24) holds. In particular, one can set $\alpha_t = t^{-\theta}$, where $\theta > 0$, and $K = cn$ for some constant $c > \frac{1}{2}$. In this case $\sum_{t=1}^{T}\alpha_t$ dominates $\sum_{t=1}^{T}\alpha_t^2$ and for any $t$:

$$\sum_{k=1}^{t}(2\gamma^2)^k\frac{\alpha_{t-k}^2}{\alpha_t} \le \sum_{k=1}^{t}\left(2\left(1 - \frac{K}{n}\right)\right)^k\frac{\alpha_{t-k}^2}{\alpha_t} = \sum_{k=1}^{t}(2 - 2c)^k\frac{t^\theta}{(t-k)^{2\theta}}$$

Since powers dominate polynomials, this sum converges in the limit as $t \to \infty$, so the condition in (24) is guaranteed to hold.

In the case when $K = cn$ with $c > \frac{1}{2}$, one can also set a fixed learning rate:

$$\alpha = \sqrt{\frac{f(x_0) - f(x^*)}{T\left(2LM^2 + 4L^2M^2\left(1 + \frac{\xi}{P\gamma}\right)^2 D\right)}}. \tag{30}$$

Then we obtain:

$$\min_{t\in\{1,\dots,T\}}\mathbf{E}\left[\|\nabla f(v_t)\|^2\right] \le \frac{1}{T}\sum_{t=1}^{T}\mathbf{E}\left[\|\nabla f(v_t)\|^2\right]$$
$$\le \frac{4\left(f(x_0) - f(x^*)\right)}{T\alpha} + \frac{\left(2LM^2 + 4L^2M^2\left(1 + \frac{\xi}{P\gamma}\right)^2 D\right)T\alpha^2}{T\alpha}$$
$$\le 5\sqrt{\frac{\left(f(x_0) - f(x^*)\right)\left(2LM^2 + 4L^2M^2\left(1 + \frac{\xi}{P\gamma}\right)^2 D\right)}{T}}.$$

## C   Experimental Details

**Datasets and models.** We evaluated the algorithm on two machine learning tasks, namely classification and linear regression. We train ResNet110 [12] on CIFAR-10 [13] for image classification. We train a linear classifier on the RCV1 corpus [14] using logistic regression and perform linear regression to train a model on a synthetic dataset containing 10K samples with 1024 features randomly generated with some Gaussian noise added.

**Setup.** We conduct experiments by implementing the algorithm into the two frameworks CNTK [32] and MPI-OPT [20]. The latter is a framework developed to run distributed optimization algorithms such as SGD or SCD on multiple compute nodes communicating via any MPI library with minimal

overhead. We make use of SparCML [20] as the communication layer to efficiently aggregate the sparse gradients. Implementation details can be found in [20]. For image classification, we use standard batch sizes and default hyper-parameters form the full accuracy convergence in all our experiments, which we define to be our baseline. These values are given in the open-source CNTK 2.0 repository. The image classification, experiments are conducted on 4 nodes. We tune the hyper-parameters such as batch-size, initial learning rate and decay factor for logistic and linear regression in order to achieve best possible convergence on the full accuracy baseline setting. We set those values for performing experiments with various values for $K$ and perform the experiments using 8 nodes.