[Reviews · NeurIPS 2018]

Reviewer 1



This paper provided the first theoretical analysis of the popular “TopK” sparsification-based communication reduction technique. The proposed approach is fairly general, which should extend to methods combining sparsification with quantization by reduced precision, and methods using approximate quantiles. Overall, the presentation of this paper is good. The related works and different research pesperctives for communication efficient distributed sgd are well overviewed and organized . While I'm not an expert in the topic of this paper, I have no idea about the originality and significance of the theoretical results and proof techniques. The main idea of the paper looks sound and good. The experiments are well designed. I think the paper is good. One minor thing, the reference 12, 13 are duplicated.

Reviewer 2



The paper proposes a theoretical study of a heuristic method for distributed SGD, using the sparsification gradients techniques. The technique is not new, but this theoretical study is, in particular authors claim that their analysis does not require a bound on the maximum delay, which departs from already known analysis for asynchronous variants of SGD. I am however not confortable with the main assumption of the paper (Assumption 1), which is not discussed enough in my opinion, or numerically checked in experiments. It is I believe a strong assumption, that provides an ad-hoc control of the TopK operator. A comparison with the bounded delay assumption for asynchronous SGD is not fair, since this assumption is usually satisfied / numerically checked in experiments, while it is hardly the case of Assumption 1 for the TopK SGD. I think that the numerical experiments are weak : the comparison with asynchronous SGD is on a convex problem only (while the paper markets the use of TopK SGD for deep nets) and only illustrate marginal improvements. As a conclusion, I find the paper interesting, and won't oppose its acceptance, but I think that it is only marginally above the acceptance threshold for the NIPS venue.

Reviewer 3



This paper proposes the first convergence guarantees for sparsification-based distributed training of massive machine learning models. It’s a solid theoretical work and the proof is not difficult to follow. If I understand the paper correctly, the Assumption 1 is the core assumption, but it’s quite abstract. It seems better to give some easy to understand “meta assumptions”, when these assumptions are satisfied, the Assumption 1 holds true, or show some specific examples, for what kind of objective function and order of magnitude, the Assumption 1 can be satisfied.